# Targeted Antisense Oligonucleotide-Mediated Skipping of Murine *Postn* Exon 17 Partially Addresses Fibrosis in D2.*mdx* Mice

**DOI:** 10.3390/ijms25116113

**Published:** 2024-06-01

**Authors:** Jessica Trundle, Ngoc Lu-Nguyen, Alberto Malerba, Linda Popplewell

**Affiliations:** 1Department of Biological Sciences, School of Life Sciences and Environment, Royal Holloway University of London, Surrey TW20 0EX, UK; j.trundle@ucl.ac.uk (J.T.); ngoc.lu-nguyen@rhul.ac.uk (N.L.-N.); 2Developmental Biology and Cancer Research and Teaching Department, University College London Great Ormond Street Institute of Child Health, London WC1N 1EH, UK; 3National Horizons Centre, Teesside University, Darlington DL1 1HG, UK

**Keywords:** Periostin, fibrosis, Duchenne muscular dystrophy, D2.*mdx*

## Abstract

Periostin, a multifunctional 90 kDa protein, plays a pivotal role in the pathogenesis of fibrosis across various tissues, including skeletal muscle. It operates within the transforming growth factor beta 1 (Tgf-β1) signalling pathway and is upregulated in fibrotic tissue. Alternative splicing of Periostin’s C-terminal region leads to six protein-coding isoforms. This study aimed to elucidate the contribution of the isoforms containing the amino acids encoded by exon 17 (e17+ Periostin) to skeletal muscle fibrosis and investigate the therapeutic potential of manipulating exon 17 splicing. We identified distinct structural differences between e17+ Periostin isoforms, affecting their interaction with key fibrotic proteins, including Tgf-β1 and integrin alpha V. In vitro mouse fibroblast experimentation confirmed the TGF-β1-induced upregulation of e17+ Periostin mRNA, mitigated by an antisense approach that induces the skipping of exon 17 of the *Postn* gene. Subsequent in vivo studies in the D2.*mdx* mouse model of Duchenne muscular dystrophy (DMD) demonstrated that our antisense treatment effectively reduced e17+ Periostin mRNA expression, which coincided with reduced full-length Periostin protein expression and collagen accumulation. The grip strength of the treated mice was rescued to the wild-type level. These results suggest a pivotal role of e17+ Periostin isoforms in the fibrotic pathology of skeletal muscle and highlight the potential of targeted exon skipping strategies as a promising therapeutic approach for mitigating fibrosis-associated complications.

## 1. Introduction

Duchenne muscular dystrophy (DMD) is a severe neuromuscular condition affecting 1:3500–1:5000 newborn boys worldwide [1,2]. Patients die in their thirties–forties, mainly due to cardiomyopathy and respiratory insufficiency [3]. Skeletal muscle fibrosis is among the most prominent pathological features of DMD. Accordingly, the levels of transforming growth factor beta 1 (Tgf-β1), a major pro-fibrotic factor, were found to be elevated in DMD patients [4], as well as in the diaphragm of the *mdx* DMD mouse model [5].

Periostin (*Postn*) is an enhancer of Tgf-β1-induced fibrosis [6], promoting Tgf-β1 synthesis and activating its signalling pathways [7,8,9]. Periostin is overexpressed in a variety of fibrotic disorders, where it acts as a promoter of collagen fibrillogenesis through its interaction with bone morphogenetic protein 1 (Bmp-1) to proteolytically activate lysyl oxidase (Lox) that catalyses collagen cross-linking [10]. In skeletal muscle, Periostin is strongly upregulated in response to injury [11,12]. Conditions such as muscular dystrophies result in a state of chronic injury, leading to skeletal muscle fibrosis. The contribution of Periostin to this process is supported by the finding that increased expression of Periostin in the δ-sarcoglycan-null mouse model of muscular dystrophy exacerbates disease pathology, while its deletion has a positive effect on the reduction of fibrosis [13]. For DMD, Periostin has been reported to be upregulated in the gastrocnemius muscle of *mdx* mice [14] and the diaphragm of *mdx-4cv* mice [15], two models of the disease. We have recently demonstrated a positive correlation of Periostin expression in the diaphragm of *mdx* mice compared to the C57BL/10 control [16], with increased fibrosis through collagen deposition.

Periostin is expressed as multiple isoforms with variable C-terminal domains [17] as a result of alternative splicing within the exon 16–23 region of the gene. Exon 17-containing *Postn* splice variants have been implicated in fibrosis in both acute myocardial infarction [18] and DMD murine models [19]. Understanding the contribution of the various Periostin isoforms to skeletal muscle fibrosis could highlight new therapeutic targets or biomarkers for this aspect of muscular dystrophy phenotypes.

Here, we provide an in silico-based foundation for this with structural modelling and an examination of predicted docking interactions of the various Periostin isoforms with fibrotic proteins, establishing that the differences seen correlate with the presence or absence of the domain encoded by exon 17. We further assessed the potential of antisense oligonucleotide-induced *Postn* exon 17 skipping as an antifibrotic therapy strategy, under the hypothesis that e17+ Periostin isoforms are involved in the fibrotic pathway of skeletal muscle, whilst e17− Periostin isoforms function otherwise. We showed that an optimised antisense oligonucleotide with phosphorodiamidate morpholino oligomer (PMO) chemistry designed to target *Postn* exon 17 effectively reduced e17+ Periostin mRNA expression in TGF-β1 stimulated mouse skeletal muscle fibroblast cells.

Systemic antisense treatment of the D2*.mdx* mouse model, characterised by extensive fibrosis in skeletal muscles [20,21], decreased both the e17+ variant and Periostin protein expression alongside a collagen deposition reduction, whilst normalising the level of grip strength to that seen in the wild type.

Given the potential of these findings, the application of antisense oligonucleotide-induced *POSTN* exon 17 skipping therapy in human DMD patients holds promise, with optimisation of the delivery approach required for improved translation. The demonstrated efficacy in reducing fibrosis and improving muscle function in the D2.*mdx* mouse model suggests that a pathology-targeting approach could similarly benefit human patients by mitigating the progression of fibrosis and preserving muscle function.

Taken together, our findings suggest that this targeted approach may hold promise as a novel strategy for mitigating the progression of fibrosis in DMD patients, offering new avenues for therapeutic intervention, and improving the clinical management of this debilitating condition.

## 2. Results

### 2.1. Bioinformatic Analysis of Periostin Variants Reveals Structural and Docking Disparities

In this investigation, we employed the Ensembl database to assess the nucleotide and amino acid sequences of the *POSTN* gene and its corresponding proteins [22]. Six protein-coding variants of Periostin were identified: POSTN-201 (ID: ENST00000379742.4), POSTN-202 (ID: ENST00000379743.8), POSTN-203 (ID: ENST00000379747.9), POSTN-204 (ID: ENST00000379749.8), POSTN-209 (ID: ENST00000541179.5), and POSTN-210 (ID: ENST00000541481.5). These variants are dissimilar in length and exon compositions, with POSTN-203 identified as the full-length isoform encoding all 23 exons [23]. The variations in sequence lay within the alternative splicing region, with exons 17 to 21 among the identified exonic variations. A bioinformatic analysis was conducted on human Periostin in order to provide a foundation for the experimental concept and justification of the subsequent in vivo experimentation. A homology of 89.2% was recorded between human and mouse counterparts [24], with exonic combinations within protein-coding isoforms being identical across species [22].

Tertiary protein structure prediction was executed using Phyre2 [25] and subsequently visualised through Jmol [26]. Divergent structural configurations were observed both within and outside of the alternatively spliced region (Figure 1A). The protein region encoded by exons 1–15 showcased structural disparities even with identical amino acid sequences. The absence of the domain encoded by exon 17 resulted in a secondary structure variation with observed differences in beta sheets and alpha helices compared to full-length Periostin within the part of the protein encoded by exons 18, 19, and 21.

ScanProsite [27] was used to identify the functional domains of the POSTN variants. An EMI domain and four FAS1 domains were identified and located within exons 1–15, as shown in Figure 1B. These domains, linked with multimer formation and cell adhesion, were consistent within all analysed variants.

Protein–protein interactions were then predicted utilising ClusPro [28] and visualised with Jmol spacefill modelling (Figure 1C). This analysis identified distinctive interaction points for e17+ isoforms compared to e17− isoforms, including notably with Tgf-β1, a known fibrosis stimulant.

### 2.2. Design of Antisense Oligonucleotides for the Targeted Skipping of Postn Exon 17

We designed three PMO candidates for m*Postn* exon 17 skipping. ESEFinder [29] was utilised to identify SR protein motifs (Figure 2A). The SR protein site analysis highlighted a clear predicted region of highly clustered ESE sites starting at approximately 3′ exonic position 46 and continuing across the exon/intron boundary. A further peak in the exonic region occurs across 5′ nucleotide positions 93–95 where two ESE sites overlap. A splice site analysis identified the predicted donor and acceptor splice sites, which were then mapped onto a predicted secondary structure, using mfold [30], of exon 17 plus 100 nt of the flanking intronic sequence seen in Figure 2B. The mfold structural results further identified areas of open structure that could be targeted with antisense oligonucleotide sequence design.

Our analysis of *Postn* exonic sequence incorporated open structure%, GC%, and G% of the target sequence (indicative of a high potential for quadruplex formation [31]), splice site location, and predicted total binding affinity. These thresholds were combined to design three PMOs with maximum binding potential and target coverage. The chosen PMOs covered 95% of the exon 17 sequence. Within the exonic region, the PMOs covered 98% of the open structure. The location and characteristics of each PMO are summarised in Figure 2B and Figure 2C, respectively.

### 2.3. TGF-β1-Induced Expression of e17+ mPostn Is Mitigated by Targeted mPostn Exon 17 Skipping In Vitro

In order to test the efficacy of these antisense oligonucleotides targeting *Postn* exon 17, the murine Mh fibroblast cell line was used. The cells were treated with 5, 10, 15, 20, and 25 ng/mL of TGF-β1 to enhance the expression of profibrotic genes and compared the effect to an untreated Mh cell negative control.

We demonstrated that when TGF-β1 is not introduced, the expression of e17− m*Postn* is elevated in comparison to e17+ m*Postn* (Figure 3A). Upon TGF-β1 stimulation, the expression of e17+ m*Postn* increased to levels higher than that of e17− transcripts. Notably, with escalating doses of TGF-β1, the proportion of m*Postn* expression attributed to e17+ transcripts exhibited a concomitant increase (Figure 3B). The lowest dose of TGF-β1 that resulted in a statistically significance increase in e17+ m*Postn* transcripts was 10 ng/mL (*p* < 0.0001). Since higher doses of TGF-β1 did not substantially increase e17+ m*Postn* expression, the 10 ng/mL TGF-β1 dose was chosen for the following experiments. We then conducted a TGF-β1 time response experiment to investigate the effect of varying the incubation time with 10 ng/mL of TGF-β1 on the expression of e17+ m*Postn* in the Mh fibroblast cell line (Figure 3B,D). Incubation times of 6, 24, 30, 48, and 72 h were considered, alongside an untreated control (0 h) serving as the baseline. We demonstrated a time-dependent increase in the expression of e17+ m*Postn*, with the most significant differences at the 48- and 72-h time points (*p* < 0.0001). Our finding confirmed the significant increase in e17+ m*Postn* expression after a minimum of 24 h of treatment with 10 ng/mL of TGF-β1 (*p* = 0.0130), and that longer incubation times elevated the expression levels further (*p* < 0.0001). We therefore decided to use 10 ng/mL of TGF-β1 and 48-h incubation to test the PMOs.

Each PMO was added at 0.5, 1, 2, and 5 μM doses, and *Postn* expression was compared to untreated unstimulated and TGF-β1-stimulated controls. The RT-qPCR analysis showed that 10 ng/mL of TGF-β1 significantly elevated the expression levels of e17+ m*Postn* RNA normalised to *Rplp0* expression compared to the negative control (*p* = 0.0086, *p* < 0.0001, *p* < 0.0001) (Figure 3E). All PMOs, at all doses, significantly reduced the TGF-β1 -induced expression of e17+ m*Postn* (*p* < 0.0001). PMO3 was identified as the lead candidate demonstrating the most effective e17+ m*Postn* reduction (*p* < 0.0001) at the lowest dose.

### 2.4. Intraperitoneal vivoPMO3 Delivery Significantly Decreases mPostn Exon 17 Expression and Improves Grip Strength Muscle Function in the D2.mdx Mouse Model

We subsequently investigated the effects of PMO3 administration in vivo using D2*.mdx* mice. Fibrosis establishment occurs between 1 and 2 months of age in this model [21]. To establish the mRNA expression profile of e17+ m*Postn* and determine the age to start the PMO treatment, we analysed 2-, 6-, and 24-week-old D2*.mdx* muscle to assess the muscle before, during, and after the establishment of fibrosis, respectively.

All muscles showed minimal e17+ m*Postn* expression at 2 weeks of age. The levels were then elevated and peaked in 6-week DIA (*p* < 0.0001), QUAD (*p* = 0.0003), GAS (*p* < 0.0001), and HRT (*p* = 0.0003), compared to 2-week muscles, and decreased at the 24-week timepoint (DIA *p* = 0.0005, QUAD *p* = 0.0007, GAS *p* = 0.0005, HRT *p* = 0.0007) (Figure 4). We observed no significant expression of e17+ m*Postn* in TA muscle at all age groups. The data indicated that a pre-6-week timepoint is likely necessary for preventive intervention with vivoPMO3 for exon 17 skipping.

We therefore treated a group of D2.*mdx* mice (N = 5) with vivoPMO3 on a weekly basis starting at 2 weeks of age. We injected another group of D2.*mdx* mice with scrambled vivoPMO (N = 6), which was considered as a negative control, and a group of Dba.2J (Wt) mice with scrambled vivoPMO (N = 4), which was considered as a positive control. vivoPMOs were delivered via intraperitoneal injection at a dose of 10 mg/kg. As the diaphragm (DIA) is particularly affected in the D2.*mdx* model and reproduces the muscular pathology observed in patients better than other muscle types, we focused our study on this respiratory muscle. DIA muscle was selected for analysis due to its strongest fibrotic response in the D2.*mdx* characterisation study (Figure 4). DIA from vivoPMO3-treated D2.*mdx* mice expressed significantly less e17+ m*Postn*-containing transcripts than the scrambled vivoPMO-treated D2.*mdx* group (*p* = 0.0021) (Figure 5A), completely normalising the expression level to the wild-type value (*p* = 0.9468). We further observed a significant decrease (*p* = 0.0383) in protein expression in the vivoPMO3-treated group (Figure 5C,D) compared to the group given the scrambled vivoPMO treatment. The reduction in Periostin protein level led to a significant decrease (*p* = 0.0313) in the hydroxyproline content (Figure 5E), indicative of the collagen level, down to the levels seen in the wild-type control (*p* = 0.5108). The vivoPMO3 treatment also restored the forelimb grip strength (Figure 5F) to wild-type values (*p* = 0.6104), whilst the fatigue resistance level was marginally, though not significantly, improved (Appendix A).

## 3. Discussion

With the hypothesis that Periostin isoforms containing the domain encoded by exon 17 are implicated in skeletal muscle fibrosis, we examined the influence of alternative splicing between exons 16 and 23 on Periostin protein isoforms’ 3D structure and their relative interaction with extracellular matrix (ECM) proteins. Our analysis suggests that alterations in the 3D structure are not limited to the alternatively spliced C-terminus but they also extend to the remaining constant region of Periostin isoforms. These structural changes have implications for the interaction of Periostin isoforms with fibrotic proteins within the ECM. While previous works on Periostin structure have primarily focused on the unstructured nature of its C-terminal region, which nonetheless participates in biological functions by binding to ECM components [32], the use of AlphaFold AI prediction software (V2) [33] has not been used to compare isoform variants. Our findings using a Phyre2 (V2.0) intensive analysis align with previous observations regarding the disordered tertiary arrangement of secondary structure elements in Periostin and extend to non-full-length isoforms. Specifically, we identified distinct alterations in beta sheet and alpha helix elements when exon 17 is absent in a transcript, highlighting the importance of exon 17-containing (e17+ Periostin) isoforms in Periostin’s structural dynamics and its potential role in fibrosis regulation.

The mechanism by which e17+ Periostin isoforms contribute to fibrosis has not been previously explored. Our examination of the differential protein folding of the isoforms suggests that changes in the conformation of variants lacking the exon 17-coding domain have the potential to alter the downstream function of the protein, such as the accessibility of integrins to their binding sites. Previous studies have also highlighted the importance of integrins in mediating the functions of Periostin [34,35]. Specifically, Periostin has been shown to interact with tumour cells through integrin receptors, contributing to processes such as cell proliferation and migration [36]. The four fasciclin-like (FAS1) domains of Periostin are known to interact with integrins αvβ3, αvβ5, and α6β4, while the C-terminal region is involved in interactions with collagen and fibronectin [37]. Several studies have demonstrated the binding of αvβ3 and αvβ5 integrins to Periostin in various cell types, including osteoblasts and cancer cells, leading to the activation of signalling pathways such as the Focal Adhesion Kinase (FAK) pathway [34].

In previous studies, the Periostin C-terminal region was associated with a network of 143 potential protein interactors, with a particular emphasis on syndecan-1 and syndecan-4, whose binding depends on the presence of the full-length C-terminal region [32]. These heparin sulphate proteoglycans are reported to interact with the heparin-binding site located within the arginine-rich motif at the distal end of the Postn C-terminal region. Notedly, syndecan-1 has been implicated in promoting fibrosis in the lung [38], whereas syndecan-4 has been reported to mitigate fibrotic processes through Tgf-β1 attenuation [39].

Whilst mechanistic conclusions are outside the scope of our current study, our bioinformatic data suggest that alternative splicing could drive structural alterations that have downstream effects on the binding to integrins, heparin sulphate proteoglycans, and other known interactants. It would be important to explore the effects of the alternative splicing of the C-terminal region with the binding capability of the protein.

Although the altered biology of Periostin splice variants has not yet been revealed, it has been suggested that the full-length isoform of the Periostin protein is rarely secreted, whereas the shorter isoforms predominate in the ECM [40]. Furthermore, by analysing the expression patterns of different isoforms in intact and regenerating muscles, Ito and colleagues suggested that the different functions of the various Periostin isoforms are required for the maintenance of muscle fibres during regeneration [41]. In addition to this, it has been demonstrated that the full-length isoform promotes the deposition of interstitial collagen in a rat myocardial infarction model, whereas the isoforms that lack the domain encoded by exon 17 promoted cardiac repair [18]. This report suggests that antisense manipulation of Periostin isoforms within cardiac tissue could facilitate improved function. Whilst we did see significant Postn changes in D2.*mdx* heart muscle, the model had previously been demonstrated to provide inconsistent readouts as a cardiac fibrosis model [42]. We believe that the investigation of Postn exon skipping in the heart in an alternative model could provide more relevant data.

The manipulation of Periostin isoforms as antifibrotic approach in skeletal muscle has not been examined before. Here, we provide evidence of a link between fibrotic onset and *Postn* splice variations in the D2.*mdx* mouse model. Periostin upregulation, with no delineation of the splice variant expressed, has previously been described in C57.*mdx* mice [14], as well as in *Sgcg*^−/−^ [13] and *mdx4-cv* [15] models. In the D2.*mdx* model, Periostin exon 17 upregulation was demonstrated compared to Dba.2J wild-type DIA [19] at 13 weeks of age. Here, we also observed an upregulation of e17+ m*Postn* expression in comparison to the Dba.2J wild type in DIA, QUAD, and GAS muscles at 6 weeks, with highest level observed in the DIA. These data challenge the findings from two key papers concerning the D2.*mdx* mouse model characterisation [20,21] that concluded that the establishment of fibrosis occurs at 2 months of age. In our study, we demonstrated an earlier fibrosis onset at the mRNA level, as evidenced by high e17+ m*Postn* expression at 6 weeks, whilst at 2 weeks, minimal expression was observed. We therefore suggest 2 weeks of age as a suitable timepoint for preventative antifibrotic treatment administration in this model. Previous studies on vivoPMO delivery in C57.*mdx* mice have primarily focused on older age groups, specifically at 9 and 10 weeks of age [43,44], with optimisation for delivery by GeneTools documented in mice as young as 4 weeks [45].

Using this timepoint, we demonstrated a mitigation of e17+ m*Postn* upregulation in D2.*mdx* via the administration of weekly IP vivoPMO designed to skip *Postn* exon 17. The reduction seen in the DIA coincided with reduced full-length Periostin protein and hydroxyproline levels. Furthermore, the treatment was associated with a significant forelimb strength increase in the treated group compared to untreated D2.*mdx* to levels consistent with the wild type. Whilst grip strength was restored, fatigue distance was not, suggesting that further optimisation of the dosing regimen and consideration of a combinational approach may be necessary to further ameliorate the disease. The combined delivery of antisense exon skipping therapies and adeno-associated viral vector-mediated gene addition to restore dystrophin expression have demonstrated an improvement in muscle function [46] and enhancement of skipping efficacy [47] compared to individual treatment. A combination treatment of a myostatin-targeting PMO (BPMO-MSTN) alongside a dystrophin restoration PMO targeting exon 23 in the DMD gene (BPMO-M23D) has previously demonstrated improvements in functional assessments [48]. Grip strength, endurance, and coordination tests, revealed significant improvements in combined BPMO-M23D&MSTN-treated mice compared to saline-injected C57.mdx mice and single PMO-treated mice, again suggesting the potential of the combinational approach to boost functional improvements.

In addition to the potential of the combinational approach, our exon-skipping strategy could benefit from recent advancements in delivery vectors for DMD therapeutics. Innovations such as nanoparticles [49] and click-chemistry conjugate synthesis [50] have shown significant promise in enhancing drug delivery and achieving specific tissue targeting [51]. While our current study employed vivoPMO dendrimer conjugation, exploring alternative delivery methods may further enhance the efficiency, specificity, and overall effectiveness of the treatment.

Improving the delivery of alternative therapeutics for DMD is crucial to mitigate the adverse side effects associated with current corticosteroid treatments. Although corticosteroid therapy, when initiated before the age of five, is linked to a delayed loss of ambulation, extended survival, and improved pulmonary function [52], its severe side effects significantly limit its therapeutic efficacy [53]. Consequently, our research aims to alleviate these adverse effects by investigating alternative phenotype-addressing therapies.

Taken together, we have provided further evidence of the presence of Periostin in the context of fibrotic pathology in skeletal muscle. We have shown a positive relationship of e17+ *Postn* splice variants with disease in a mouse model of DMD, which exhibits progressive fibrosis development [21], thus highlighting the potential of these variants as drivers of fibrosis. Our data support the hypothesis that there is a switch in Periostin expression from non-fibrotic variants to a pro-fibrotic expression pattern with disease, which is also triggered by an increase in TGF-β1 fibrotic stimulation in mouse fibroblasts.

Our data provide further insights into the mechanism of disease progression in DMD and establishes a pro-fibrotic role for exon 17-containing splice variants of Periostin in the establishment of skeletal muscle fibrosis. This has potential implications for the development of improved and more targeted approaches in the treatment of this pathological phenotype of DMD, as well as other muscular dystrophies.

## 4. Materials and Methods

### 4.1. Structural Analysis of Periostin Protein Products

The Ensembl genome browser [22] was used throughout the analysis of *POSTN* (ID: ENSG00000133110) for the retrieval of the gene and protein sequences. Of the 10 splice variants of *POSTN* listed, 6 were categorised as protein coding (transcript *POSTN-201* (ID: ENST00000379742.4), transcript *POSTN-202* (ID: ENST00000379743.8), *POSTN-203* (ID: ENST00000379747.9), transcript *POSTN-204* (ID: ENST00000379749.8), transcript *POSTN-209* (ID: ENST00000541179.5), transcript *POSTN-210* (ID: ENST00000541481.5)) and were analysed further. *POSTN-203* encodes the full-length Periostin protein (exons 1 to 23), while *POSTN-201* lacks exons 17 and 18, *POSTN-202* lacks exon 17, *POSTN-204* lacks exon 21, *POSTN-209* lacks exons 17 and 21, and *POSTN-210* lacks exons 17 to 19

The ScanProsite tool [27] was used to predict the domains and motifs of each Periostin protein variant. The amino acid protein sequences were submitted in turn to the tool, scanned against the PROSITE collection of motifs (option 1), and run at high sensitivity. The output selected was “Graphical View” and “retrieve complete sequences”. Profile hits of EMI and FAS1 domains were recorded as a protein position and summarised graphically. The domains were mapped onto the Protein Data Bank (PDB) 3D structures of each variant using Java molecular (Jmol) [26].

Phyre2 software (V2.0) [25] was used to predict the tertiary structure of each isoform of human Periostin protein. The primary amino acid sequence for each isoform was submitted and run on ‘Intensive’ modelling mode. Intensive modelling mode included the examination of an extensive list of hits selected for the submitted template with maximum sequence coverage and confidence. The high confidence templates were converted into 3D models and compiled in the Phyre2 ‘ab initio’ and multi-template modelling tool, Poing. This system pools together the known templates, as well as predicted structures of the protein regions lacking a previously determined template. These unknown structural regions are predicted using distance constraints and the ‘ab initio’ physics model. The model generated by the Poing system was then combined with the originally predicted template in order to give a complete structure prediction model. The structure with the highest confidence rating was opened using Java molecular (Jmol) [26], a modelling software which allows for interactivity and visual coding of the structure.

ClusPro 2.0 [28] is a protein docking tool that predicts the 3D structure of a protein complex from the input of two structural protein data banks (PDBs). The tool outputs the top 10 predicted models defined by ‘centres of highly populated clusters of low energy docked structures’. The PDB structures of each POSTN variant [25] were in turn submitted alongside the structures of TGF-β1, ITGαV, Decorin, Col1a1, FN1, IL3, and IL13 [54]. The docking model with the highest confidence was selected for each Periostin variant, and the structures were visualised using Jmol [26].

### 4.2. Design of the Antisense Sequences to Skip Postn Exon 17

Potential antisense oligonucleotide targeting locations were identified along the entire inputted exonic and intronic sequence using Sequence Manipulation Suite [55]. The Split FASTA tool segmented the full sequence into 254 potential oligo 28 mer binding sites. The sequences were reverse transcribed in the software in order to give a list of all potential AONs for the sequence.

SR protein sites were predicted using ESEFinder [29]. For the SR protein analysis, SRSF1 (red), SRSF1 IgM-BRCA1 (pink), SRSF2 (blue), SRSF5 (green) and SRSF6 (yellow) sites were analysed at thresholds of 1.956, 1.867, 2.383, 2.67 and 2.676, respectively. The secondary pre-mRNA structure of exon 17 was predicted using the online software Mfold (V3) [30]. The sequence was inputted in FASTA format and standard parameters were selected. RNA fold was selected to give a myriad of circular structure plots. The structure predicted to have the most negative energy, and thus the strongest binding energy and highest stability, was selected. The structure was visually analysed for an open loop structure within the target region.

Sfold software (V2.2) [56] was used to predict the target binding energy along the sequence, and RNAup Server [57] was used to predict interactions between the oligo and target sequence. The total free energy of binding was calculated as the energy from duplex formation minus the combined total of the opening energy for the long sequence and opening energy for the short sequence.

Regions of interest for AON selection were defined using the above analysis and subsequently narrowed to three lead candidates using the following thresholds: 0 open loop ends and >65% open structure in the target region secondary structure, as well as a predicted binding energy < −30 kcal/mol. The selected AONs give 95% coverage of exon 17, and 98% coverage over open secondary structures.

### 4.3. Cell Culture Experiments in Mh Mouse Fibroblast Cell Line

The Mh mouse fibroblast cell line was used for in vitro experimentation (CRL-2709, ATCC, Teddington, UK). The cells were cultured in full growth media, consisting of 10% foetal calf serum added to DMEM/F-12, GlutaMAX™ supplement (Gibco Life technologies, Grand Island, NY, USA), and incubated at 37 °C plus 5% carbon dioxide (CO_2_). The optimal dosage and temporal TGF-β1 fibrotic stimulation were assessed. For testing the dose response, 3 × 10^5^ cells in a well of a 6-well plate were stimulated with human recombinant (r)TGF-β1 (ab50036; Abcam, CA, USA) at 0, 5, 10, 15, 20, and 25 ng/mL doses, when they were 80% confluent. RNA extraction (RNEasy mini kit, QIAGEN, UK) and *Postn* exon 17 expression levels were assessed after 24 h. For testing the time response, 3 × 10^5^ cells were plated into 6 separate 6-well plates, treated with 10 ng/mL of Tgf-β1, and designated a unique post treatment time interval (0, 6, 24, 30, 48, and 72 h), after which, the RNA was extracted as described previously.

For testing the PMO response, 3 × 10^5^ cells per well were plated into 18 separate 6-well plates. Each plate was designated a treatment condition for the cells: PMO1 (TTTGGTTATAATTTTAGTTGCTGAAAAC), PMO2 (CTTGAATGACTTTAATTTTTGGTTCCAC), or PMO3 (CTTCCGTTTTGATAATAGGCTGAAGACT), each at doses 0, 0.25, 0.5, 1, 2, and 5μM (GeneTools, Philomath, OR, USA). Three repeats per condition were used. At 80% cell confluency, the medium was removed, and the desired PMO was added to each well alongside 6 μL Endoporter (GeneTools, Philomath, OR, USA) and media (Gibco Life technologies, Grand Island, NY, USA) to make a total volume of 1 mL per well. RNA extractions were carried out 48 h post treatment.

### 4.4. Animals and In Vivo Experimental Design

All animal procedures were performed in accordance with UK government regulations and were approved by the UK Home Office under Project License P36A9994E. Ethical and operational permission for the in vivo experiments was granted by the Animal Welfare Committee of Royal Holloway University of London. D2-mdx (IMSR_JAX:013141) and Dba-2J (IMSR_JAX:000671) mice (The Jackson Laboratory, Bar Harbor, ME, USA) were maintained in a standard 12-h light/dark cycle with free access to food and water. Samples were collected according to the TREAT-NMD protocol DMD_M.1.2.007. Male and female D2-*mdx* and Dba-2J mice were used throughout this study. The characterisation study had an N number of 5 per each age group at 2, 6, and 24 weeks old.

For the 10-week delivery experiment, vivoPMO chemistry was selected [58].

Mice were assigned into treatment groups of Dba-2J + vivoPMO scramble (N = 4), D2-*mdx* + vivoPMO scramble (N = 6), and D2-*mdx* + vivoPMO.Postn (N = 5); they were evenly distributed by gender and average bodyweight to keep the volume by bodyweight dosage consistent. Both *Postn* exon 17 skip vivoPMO (Seq 5′-3′: CTTCCGTTTTGATAATAGGCTGAAGACT) and vivoPMO scramble (Seq 5′-3′: CCTCTTACCTCAGTTACAATTTATA) (GeneTools, Philomath, OR, USA) were administered weekly via intraperitoneal systemic delivery. For the 10 mg/kg dosage, the injection volume was calculated and standardised as 4 mL/kg of bodyweight.

### 4.5. Fatigue Resistance Analysis

In the investigation of fatigue resistance, a Treadmill Simplex II apparatus (Columbus Instrumentation, Columbus, OH, USA) featuring a 15% incline was utilised. A period of 5 min was allotted for the mice to acclimatise to the equipment before the commencement of the assessment. Subsequently, the treadmill was initiated at an initial velocity of 5 m/min for the initial 5 min, following which, the velocity was augmented by 0.5 m every 1 min. The determination of exhaustion in the animals was predicated upon their inability to evade a stopper placed on the treadmill for a duration of 10 s. Distance was calculated using the time to fatigue recorded.

### 4.6. Forelimb Grip Strength Analysis

Forelimb grip strength was measured as the maximum force (g) using Linton Instrumentation (Diss, Norfolk, UK). Five discrete measurements were conducted per mouse, interspersed with a 30-s intermission between each reading. The results were then normalised to the bodyweight recorded on the assessment.

### 4.7. Sample Collection and Processing

Mice were euthanised according to schedule 1 procedures. For each mouse, the diaphragm (DIA), quadricep (QUAD), gastrocnemius (GAS), heart (HRT), and tibialis anterior (TA) were collected as required for each experiment and snap-frozen in liquid nitrogen for hydroxyproline analysis, or protein or RNA extraction (as described below). All samples were stored at −80 °C.

### 4.8. RNA Extraction from Tissues, cDNA Synthesis, and Quantitative Polymerase Chain Reaction

RNA extraction was performed using the RNeasy fibrous tissue mini kit (QIAGEN, Manchester, UK) following the protocol set out by the manufacturer. The concentration of the RNA was determined using a Nanodrop ND-1000 spectrophotometer. The OD260/280 ratio given by the spectrophotometer assessed the purity of nucleic acids, with pure nucleic acids having an OD260/280 ratio between 1.8 and 2.0.

cDNA synthesis was performed using the QuantiTect Reverse Transcription Kit (QIAGEN, Manchester, UK). Approximately 1000 ng of RNA was mixed with 2 µL of genomic DNA wipeout buffer and brought up to a total of 14 µL with water. This was incubated for 2 min at 42 °C. This was then added to 4 µL of 5× Quantiscript Reverse Transcription buffer, 1 µL of primer mix, and 1 µL of Quantiscript Reverse Transcriptase, to give a final volume of 20 µL per reaction. This reaction mix was then incubated for 15 min at 42 °C followed by 3 min at 95 °C.

For end-point polymerase chain reaction (PCR), PCR amplification was achieved using the GoTaq DNA Polymerase system (Promega, Madison, WI, USA). Approximately 100–500 ng of cDNA was used in a master mix containing 1× GoTaq reaction buffer, 1.5 mM MgCl_2_, 0.2 mM of each dNTP, 0.1 µM of each primer, GoTaq DNA Polymerase, and DEPC-treated nuclease-free water (Life Technologies, Bleiswijk, The Netherlands). The PCR conditions were as follows; initial denaturation at 95 °C for 5 min; 30 to 35 cycles of 30 s at 95 °C (denaturation), 57–60 °C (annealing), and 72 °C (extension); followed by a final 10-min extension at 72 °C. Initial temperature gradient PCRs were set up for all primer sets to find the optimal annealing temperatures. The primers were designed to target Ex16 forward and Ex18 reverse (semi-nested), resulting in the amplification of two products: e17+ (200 bp) and e17− (119 bp) *Postn* isoforms (For1: CAGTTGGAAATGATCAGCTCTTG; For2: AGTTTGTTCGTGGCAGCA; Rev: CACCGTTTCGCCTTCTTTAATC).

Resolution of the products from end-point PCR was achieved by agarose gel electrophoresis using 1× TAE buffer (diluted from 50× TAE: 242 g of tris base (Sigma-Aldrich, Gillingham, UK) in double-distilled H_2_O, 57.1 mL of glacial acetic acid (Sigma-Aldrich, Gillingham, UK), and 100 mL of a 0.5 M EDTA solution (pH 8.0) (Thermo Fisher Scientific, Waltham, MA, USA)). The gels were stained with SYBR Safe Gel Stain (Life Technologies, Bleiswijk, The Netherlands) at a ratio of 1:10,000. Electrophoresis was performed at 80 to 150 V for 30 to 90 min. The gels were examined and photographed under a UV light using a gel imager (E-Box, Vilber, Collegien, France), and quantified using GelAnalyzer [59].

Quantitative polymerase chain reaction (qPCR) was performed using the LightCycler480 system (Roche, Welwyn Garden City, UK) using 384-well plates. Syber green master mix with rgw respective primer pairs was added to the cDNA, giving a total volume of 10 μL per sample, which was analysed in triplicate. The data were analysed using the LightCycler480 software (V1.5) (Roche, Welwyn Garden City, UK) with the following conditions: activation at 95 °C (5 min); 45 cycles of 95 °C, 57 °C, then 72 °C (15 s each); and a melt curve of 95 °C (5 s), 65 °C (60 s), and 97 °C (end). For relative quantification of the genes of interest, samples were normalised to the mRNA levels of ribosomal protein, large, P0 (*Rplp0*). Primers were purchased from IDT: Rplp0 (For: TTATAACCCTGAAGTGCTCGA; Rev: CGCTTGTACCCATTGATGATG), P17 (For: ATAACCAAAGTCGTGGAACCAA; Rev: CTTCCGTTTTGATAATAGGCTGAA).

### 4.9. Protein Extraction and Western Blot

Protein extraction from tissue was performed using a 3 mm Tungsten carbide bead (QIAGEN, Manchester, UK) placed in an Eppendorf tube containing 30 mg of tissue sample in 300 µL of RLT lysis buffer, and homogenised using a TissueLyser II (QIAGEN, Manchester, UK) at 25 Hz for 4 min with rotation of tubes after 2 min. The samples were centrifuged at 14,000× *g* for 10 min at 4 °C. The supernatant (protein extract) was decanted into a fresh pre-chilled Eppendorf tube and the pellet was discarded. The total protein content of sample was measured using the DC Protein Assay (BIO-RAD, Watford, UK) following the manufacturers’ standard protocol.

For Western blotting, 10 to 25 µg of total protein was mixed with 2 µL of reducing agent (Li-Cor, Lincoln, NE, USA) and 5 µL of 4× lithium dodecyl sulphate (Li-Cor, Lincoln, NE, USA) and brought up to a final volume of 20 µL with water. The samples were denatured at 70 °C for 10 min. MOPS running buffer (Li-Cor, Lincoln, NE, USA) and 4–12% Bis-Tris gels (Li-Cor, Lincoln, NE, USA) and 0.45 µm nitrocellulose membranes (Amersham, GE healthcare, Braunschweig, Germany) were used. For protein detection, goat anti-periostin/OSF-2 isoform 2 antibody (AF2955; R&D Biosystems, Minneapolis, MN, USA) at 1:2000 and the housekeeping mouse anti-vinculin antibody (Sigma-Aldrich, Gillingham, UK, SAB4200080) at 1:5000 were used. Secondary antibodies (anti-goat-Ab (green/800), anti-rabbit-Ab (red/680)) were obtained from Li-Cor. Membranes were visualised using the Odyssey CLx system (Li-Cor, Lincoln, NE, USA) and analysed using the system software Image Studio lite 5.2.

### 4.10. Hydroxyproline Quantification

Quantification of hydroxyproline present in DIA muscle samples was conducted using the Sigma-Aldrich Hydroxyproline Assay Kit (MAK357). The tissue was first homogenised in ultrapure water, added in equal amounts to 10 M concentrated NaOH, and heated at 100 °C for 2 h. After cooling on ice, the samples were neutralised using 10 M concentrated HCL, vortexed, and centrifuged. Approximately 10 µL of each neutralised sample hydrolysate was evaporated in a clear, flat-bottomed 96-well plate in duplicate, alongside prepared hydroxyproline standards, at 65 °C. A 100 µL volume of oxidation reagent mix was added to each well and the plate was incubated at room temperature for 20 min. A 50 µL volume of developer solution was added and the plate was incubated at 37 °C for 5 min, before 50 µL of a DMAB concentrate solution was added and the plate was incubated at 65 °C for 45 min. Absorbance at A560 endpoint was read and hydroxyproline quantification was achieved via comparison to the standard curve.

### 4.11. Statistical Analysis

The statistical analysis was performed using one-way ANOVA with Tukey post hoc tests to compare the wild-type and D2.*mdx* groups as well as the different treatments using the GraphPad Prism software 9.0, with *p* values < 0.05 considered statistically significant (* = *p* < 0.05, ** = *p* < 0.01, *** = *p* < 0.001, **** = *p* < 0.0001). Any outliers were identified using the ROUT method (Q = 1%), and if any were removed, they are mentioned in the respective figure legend.

## Figures and Tables

**Figure 1 ijms-25-06113-f001:**
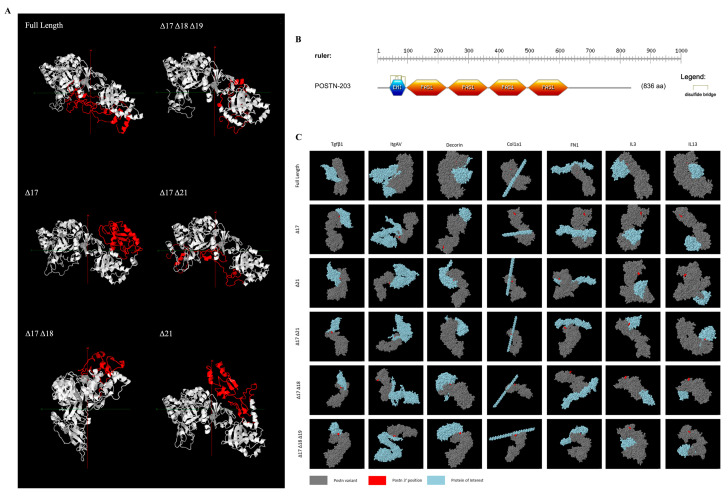
**Bioinformatic analysis of Periostin isoforms.** (**A**) Phyre2 [25]-predicted tertiary structure highlighting the C-terminal region (red) for each protein-coding isoform. (**B**) Domain and motif prediction using ScanProsite [27]. (**C**) Protein–protein docking prediction of each of the 6 protein-coding Postn isoforms with interactants Tgf-β1, integrin alpha-5 (Itgαv), decorin, procollagen (col1α1), fibronectin (FN1), interleukin 3 (IL3), and interleukin 13 (IL13). For ease of complex orientation visualisation, the N-termini of each Periostin isoform (grey) is labelled in red, with the interactant in blue.

**Figure 2 ijms-25-06113-f002:**
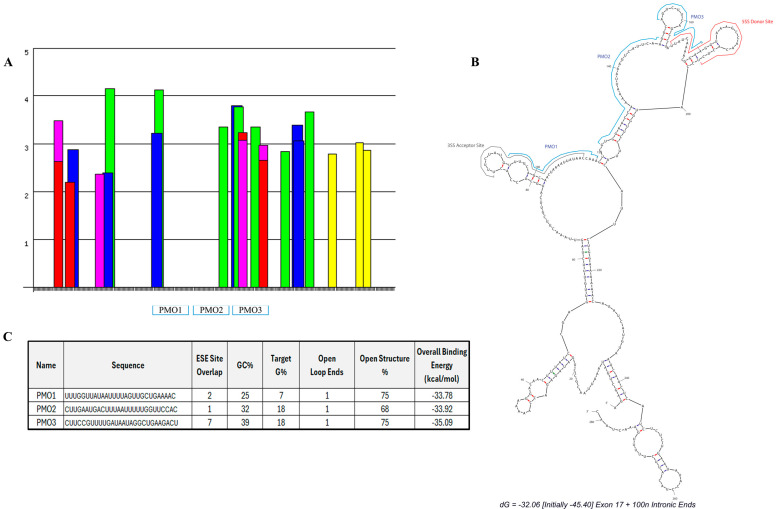
**Designed antisense oligonucleotides for m*Postn* exon 17 skipping.** SR protein-binding sites (**A**) were identified using ESEfinder [29]. SRSF1 (red), SRSF1 IgM-BRCA1 (pink), SRSF2 (blue), SRSF5 (green), and SRSF6 (yellow) sites were analysed at thresholds of 1.956, 1.867, 2.383, 2.67, and 2.676, respectively. (**B**) Lead PMO candidates and splice sites (ESEfinder) were mapped onto the mfold secondary structure prediction [30]. (**C**) Sequence and characteristics of each designed PMO.

**Figure 3 ijms-25-06113-f003:**
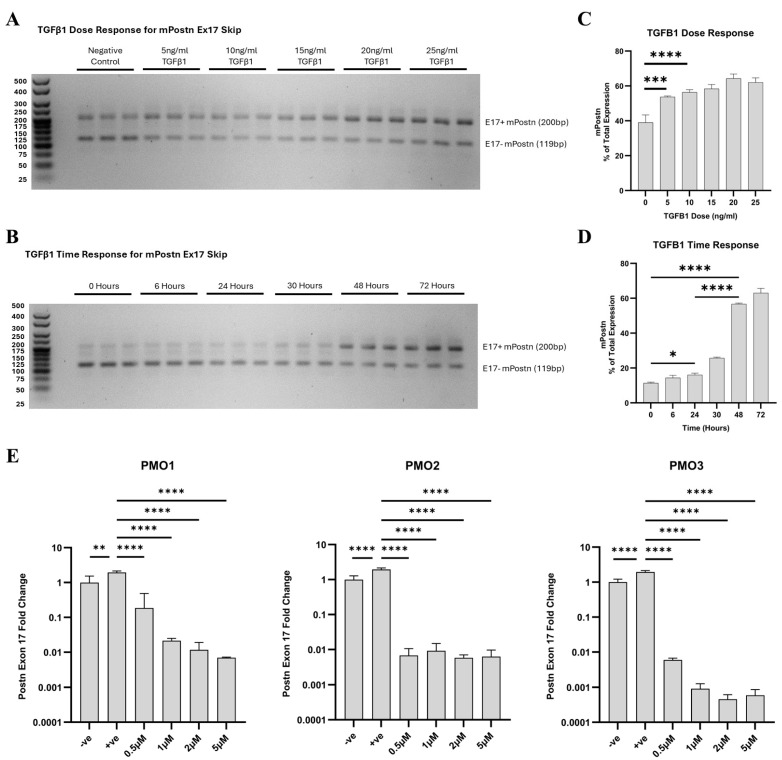
**In vitro optimisation of *Postn* exon 17 skipping antisense PMOs.** TGF-β1 dose (**A**,**C**) and time response (**B**,**D**) PCR products were run on 3% agarose gels, with the longer product containing *Postn* exon 17 (*Ex17+ mPostn*) and the shorter one lacking exon 17 (*Ex17*− *mPostn*). The *Ex17 + mPostn* band pixel density was partially quantified on a GelAnalyzer and normalised to total band expression to give a percentage readout. (**E**) RT-qPCR results of fold change in *Postn* exon 17 expression in response to PMO1–3 in vitro. All treatment groups, at all doses, were significantly reduced compared to the +ve control. Negative control (−ve) = unstimulated Mh cells; positive control (+ve) = 10 ng/mL TGF-β1-stimulated Mh cells; and treatment groups = 10 ng/mL TGF-β1-stimulated Mh cells plus designated PMO1–3 dose treatment (0.5 µM, 1 µM, 2 µM, and 5 µM). Significance (one-way ANOVA with Tukey post hoc test) recorded as * = *p* < 0.05; ** = *p* < 0.01, *** = *p* < 0.001; **** = *p* < 0.0001. Error bars = +/−SD. *N* = 3; qPCR was performed in triplicate.

**Figure 4 ijms-25-06113-f004:**
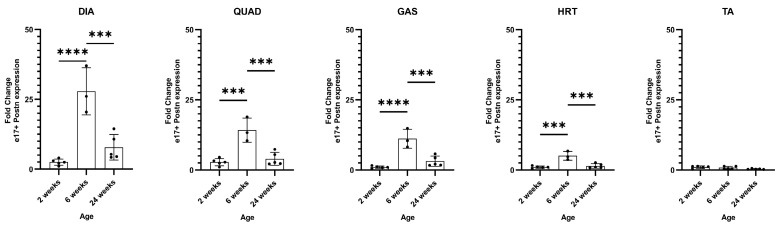
**e17+ *Postn* mRNA expression in D2.*mdx* muscle.** RT-qPCR characterisation of e17+ *Postn* expression in the DIA (diaphragm), QUAD (quadricep), GAS (gastrocnemius), HRT (heart), and TA (tibialis anterior) in D2.*mdx* mice at 2, 6, and 24 weeks of age (N = 5). Outliers were identified via the ROUT method and omitted from the statistical analysis. The data were analysed by one-way ANOVA with Tukey post hoc tests and significance was recorded as *** = *p* < 0.001; **** = *p* < 0.0001. Error bars = +/−SD; qPCR was performed in triplicate.

**Figure 5 ijms-25-06113-f005:**
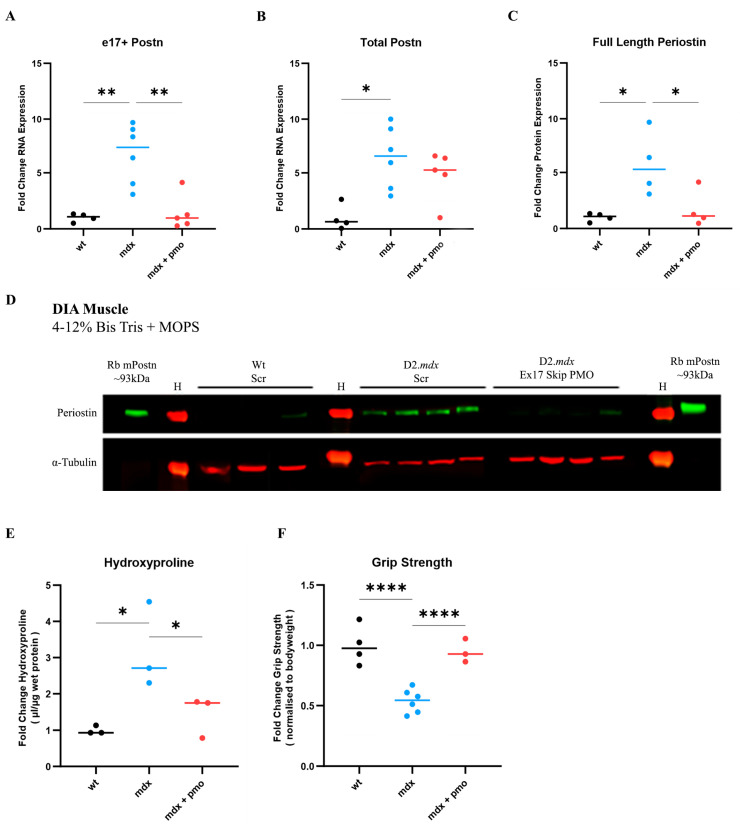
**vivoPMO significantly reduced m*Postn* exon 17 expression and induced beneficial effect on muscles.** RT-qPCR analysis of e17+ (**A**) and total (**B**) *Postn* mRNA expression represented as fold change compared to wild type (N = 4–6). (**C**) Full-length protein expression assessment using Western blot (**D**) analysis. (**E**) DIA hydroxyproline analysis (indicative of collagen level). (**F**) Forelimb grip strength was measured as the average of 5 discrete measurements of maximum force normalised to body weight. Outliers were identified via the ROUT method and omitted from the statistical analysis. The data were analysed by one-way ANOVA with Tukey post hoc tests and significance was recorded as * = *p* < 0.05; ** = *p* < 0.01; **** = *p* < 0.0001. Error bars = +/−SD; qPCR was performed in triplicate. Legend: ‘wt’ = Dba.2J wild type; ‘mdx’ = D2.*mdx* + Scramble vivoPMO; ‘mdx + pmo’ = D2.*mdx* + Postn exon 17 skipping vivoPMO.

## Data Availability

The data presented in this study are available on request from the corresponding author. The data are not publicly available due to further development of the work.

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
