# Peer review of "Targeted Antisense Oligonucleotide-Mediated Skipping of Murine Postn Exon 17 Partially Addresses Fibrosis in D2.mdx Mice"

_ijms, 2024, doi:10.3390/ijms25116113_

Round 1

Reviewer 1 Report

Comments and Suggestions for Authors

The manuscript of Trundle et al. focuses on the role of exon 17 of periostin and its impact on muscle fibrosis and function. Periostin interacts through the transforming growth factor beta 1 (Tgf-β1) signaling pathway and is upregulated after injury. Authors hypothesized that e17+ Periostin isoforms are involved in the fibrotic pathway of skeletal muscle and teh e17- Periostin isoform alleviates fibrosis. Authors showed on the fibroblasts and on the in vivo D2.mdx mouse model both on the molecular level TGF-β1-induced upregulation of e17+ Periostin mRNA, mitigated by an anti- 24 sense approach inducing skipping of exon 17. On the phenotypical level, the grip strength in the treatd mice was on the wild type level.

The introduction provides an extensive and understandable background of Duchenne muscular dystrophy, the molecular function of periostin, and its isoforms. Results are presented clearly and are divided into the bioinformatic analysis of periostin isoforms, design of the antinucleotide therapy, effects of the form with and without exon 17 on the TGF-β1 in vitro and effects of vivoPMO3 delivery in vivo in the murine model assessed by a grip strength test. The limitations are stated clearly.

The work is innovative and focuses on the not-yet well-investigated 3D function of the periostin. The mensuript is well written and deserves a publication.

Comments: 

1. Would it be possible to comment on the potential of the potential of this therapy in human DMD patients?

2. Please put this therapy context of the current landscape of the DMD treatments

3. Are any photos of the murine model muscles after treatment and controls available (if yes,  they could be added to the supplement)

4. Figure 3 images A and B could be larger for clarity

Reviewer 2 Report

Comments and Suggestions for Authors

Reviewer report on Manuscript ‘Targeted Antisense Oligonucleotide-Mediated Skipping of Murine Postn Exon 17 Partially Addresses Fibrosis in D2.mdx Mice’

In this manuscript authors aimed to elucidate the contribution of isoforms containing the amino acids encoded by exon 17 (e17+ Periostin) to skeletal muscle fibrosis and investigate the therapeutic potential of manipulating exon 17 splicing. We identified distinct structural differences between e17+ Periostin isoforms, affecting their interaction with key fibrotic proteins, including Tgf-β1 and integrin alpha V. Subsequent in vivo studies in the D2.mdx mouse model of Duchenne muscular dystrophy (DMD) demonstrated that our antisense treatment effectively reduced e17+ Periostin mRNA expression, which coincided with reduced fulllength Periostin protein expression and collagen accumulation.

Findings suggest that this targeted approach may hold promise as a novel strategy for mitigating the progression of fibrosis in DMD patients, offering new avenues for therapeutic intervention and improving the clinical management of this debilitating condition.

This article is well-designed, well-illustrated and is very interesting, from the point of view of biomedicine. The research is in the scope of the journal. Therefore, the manuscript eventually can be published after some additional minor corrections and improvements:

Introduction could be advanced. Introduction and Discussion parts of the manuscript could be advanced by overviewing some recent reviews methods: possible ways to manage Duchenne muscular dystrophy (Development of essential oil delivery systems by ‘click chemistry’ methods: possible ways to manage Duchenne muscular dystrophy. Materials, 2023, 16, 6537. https://doi.org/10.3390/ma16196537).

Conclusions could be elaborated, advanced and improved, and some additional insights on further developments and application of various nanomaterials in drug delivery systems could be added.

Comments on the Quality of English Language

 Minor editing of English language required.
